# Endplate Design and Topology Optimization of Fuel Cell Stack Clamped with Bolts

Zhiming Zhang *, Jun Zhang and Tong Zhang

School of Automotive Studies, Tongji University, Shanghai 200070, China; 2031627@tongji.edu.cn (J.Z.); tzhang@tongji.edu.cn (T.Z.)
* Correspondence: zhangzm@tongji.edu.cn; Tel.: +86-21-6958-9238

**Abstract:** The endplate plays an important role in the performance and durability of fuel cell stacks, and also to mass power density. Aiming at a lightweight endplate and uniform deflection of the endplate, the purpose of this study is to model the endplate including the supply, discharge ports and the distribution manifolds. The stress and displacement distribution of the endplate are also analyzed by numerical simulation. After that, the three optimized topologies aiming to minimize compliance, uniform stress distribution and two objectives coupling are discussed, and the intake endplate and blind endplate are individually reconstructed. The mass of optimized intake endplate is reduced by 35%, and the mass of optimized blind endplate is reduced by 46% for the goal of attaining a lightweight endplate, while maintaining the uniformity of the stress distribution of the first cell next to the endplate. Considering these factors, optimized endplates are obtained, which are valuable to fuel cell stack design.

**Keywords:** fuel cell; lightweight endplate design; topology optimization; stress distribution

## 1. Introduction

The Proton Exchange Membrane Fuel Cell (PEMFC) is a promising power source since it is friendly to the environment. It can directly convert chemical energy into electrical energy with high efficiency. Generally, hydrogen is used as the raw material, and its oxide is only water [1–3]. The development and application of fuel cells are very conductive to the realization of carbon peaking and carbon neutrality.

Since the output voltage of the single cell of the PEMFC is limited, many single cells need to be assembled and connected in series, with current collector plates, insulating plates, and endplates on both sides of the fuel cell stack in practical application. The endplates and the internal cells are compressed together to form a stack. The clamping force generated by the fasteners compresses the components of the fuel cell stack, forming a suitable contact pressure on the membrane electrodes assembly (MEA) to ensure high-efficiency operation for the fuel cell stacks [4].

The clamping force has a very important influence on the uniformity inside the fuel cell [5,6]. If the clamping force is too small, the compression of the gas diffusion layer (GDL) decreases, resulting in a sharp rise in local contact resistance, increasing electron transport resistance and an increase in local temperature [7,8]. If the clamping force is too large, the GDL is over-compressed with the porosity reduced. The gas transport resistance is increased and the MEAs may be damaged, thereby affecting the overall performance of the fuel cell stack [9–11].

At present, the design of the endplate has become a key factor affecting the contact pressure distribution between components of the fuel cell stack [12]. Increasing the thickness of the endplates can effectively improve the internal uniformity of contact pressure [13], but it will also reduce the volume power density of the whole stack, which is important to the allowable installation for a passenger car. Therefore, it is necessary to properly design the endplate to ensure a certain rigidity while reducing the mass.

A lot of improvements have been obtained in this field. Asghari et al. [14] use the finite element method to analyze the influence of the thickness of the endplate on the deformation of the bipolar plates of the fuel cell stack. In this model, the optimal thickness is 35 mm, and the internal resistance of a fuel cell stack with 5 kW is tested under different clamping torques. Carral et al. [15] studied the deformation of the endplate under the clamping force by experiment. Strain was used to measure the different parts of the endplate. One formula was developed to fit the curve.

Liu et al. [16] studied the influence of different endplate materials on the deformation and endplate weight, and further extended it to different optimized endplate structures. Alizadeh et al. [17] simplified the numerical simulation by establishing a two-dimensional finite element model to study the influence of the different materials, thickness, numbers of single cells, and sealant characteristics on the stress distribution. The accuracy of the simulation was verified by the pressure film experiment. Uzundurukan et al. [18] carried out the simulation and experiment of both bolt tightening and pressure plate tightening by converting the bolt torque into the pressure on the pressure plate. The results showed that pressure plate tightening is better in improving the power of the fuel cell stack, and the uniform distributed pressure is beneficial to avoid excessive deformation of the MEAs, thereby ensuring the porosity of the GDLs.

Zhang et al. [19] designed a downwardly inclined endplate according to the deformation characteristics of the fuel cell stack with metal bipolar plates, and optimized the dimensional size of the endplates. After optimization, the uniformity of the internal contact pressure distribution of the fuel cell stack was also improved. Liu et al. [20] proposed a new pneumatic clamping device of the endplates whereby uniform pressure was applied onto the endplate by nitrogen gas. It was found that compared with the traditional clamping bolt, the new device can effectively improve the uniformity of the internal stress distribution. Tests also showed that the new device can effectively reduce the internal ohmic impedance by 18%. Alizadeh et al. [21] also proposed a similar method to obtain a more uniform pressure distribution.

Qiu et al. [22] established an intelligent method to optimize the uniformity of pressure distribution inside the fuel cell stack. By neural network learning, it can greatly simplify the limits of simulation and provide a novel idea. However, the trained model had certain shortcomings with not much data. It can only be used for a specific fuel cell stack.

Lin et al. [23] used the variable density method and the solid isotropic penalized microstructure model to perform multi-objective topology optimization on the endplate, and obtained good stress distribution uniformity. Jiang [24] used an endplate with a semi-circular cross-section. Taking the minimum strain energy of the endplate and the minimum root mean square displacement of the contact surface as the optimization objectives, the multi-objective topology optimization of the endplate was carried out for a light weight. Wei [25] studied the effect of compression on GDL through finite element analysis, and used simulated annealing algorithm to optimize the shape and topology of the endplate. Yang et al. [26] obtained the three topology structures of lightweight endplates through numerical simulation, and compared them with those before optimization. The results showed that the optimized endplates can reduce the mass of the endplates and improve the consistency of the internal pressure.

In these analysis and optimization results, the endplate models used are relatively ideal, and the influence of factors such as supply and discharge ports and manifolds on the endplate were ignored. In fact, due to the existence of supply and discharge ports and manifolds, there will be certain differences in the stress and topology of the endplate. Therefore, the purpose of this paper is to design an endplate with supply and discharge ports and distribution manifolds. Numerical simulations are performed after modeling. The stress and deformation of the endplate are analyzed. Then, topology optimization is carried out for both the intake endplate and the blind endplate. The endplates are reconstructed according to the results of topology optimization. The same simulations are then performed for the reconstructed endplates. Finally, the comparison between the

original endplate and the reconstructed is performed. The reconstructed endplates can effectively reduce the mass and keep certain stiffness, which, in result, could increase the mass power density of fuel cell stacks.

## 2. Endplate Design, Materials and Methodology

On the basis of related fuel cell endplate structure design, the intake endplate model is proposed as following. Figure 1 is an external side view of the endplate, Figure 2 is an internal side view of the endplate, and Figure 3 is a two−dimensional schematic view of the endplate. The blind endplate is the same as the intake endplate except that there are no supply and discharge ports and manifolds.

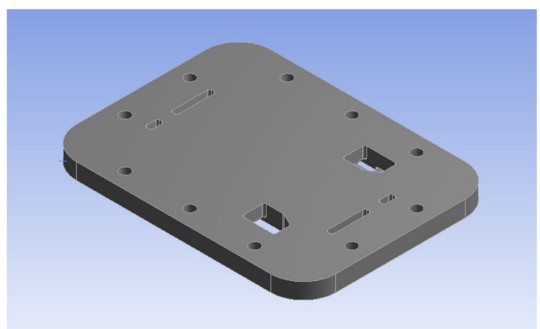

**Figure 1.** External side view of intake endplate model.

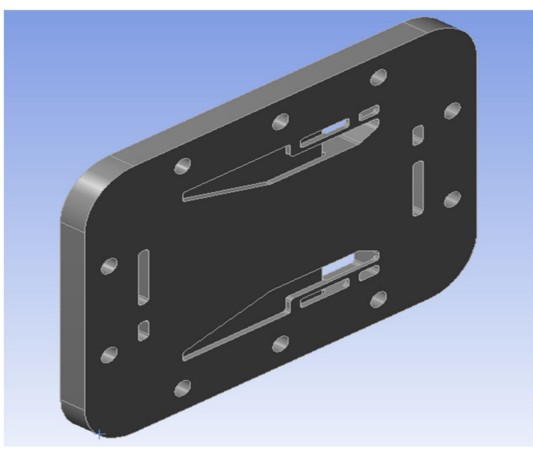

**Figure 2.** Internal side view of intake endplate model.

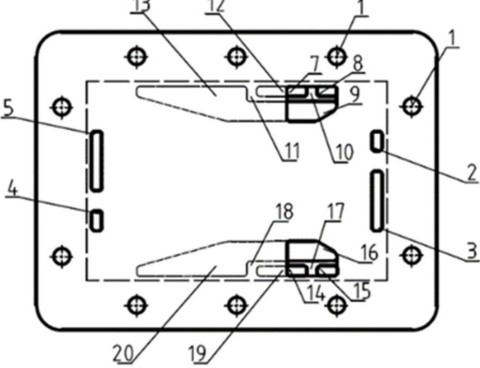

**Figure 3.** Two−dimensional schematic diagram of intake endplate: 1: clamping bolt hole; 2: anode gas supply port; 3: cooling water supply port; 4: anode gas discharge port; 5: cooling water discharge port; 7,8,9: cathode gas supply ports; 10,11: partition ribs of cathode gas supply ports; 12,13: manifolds of cathode gas supply ports; 14,15,16: cathode gas discharge ports; 17,18: partition ribs of cathode gas discharge ports; 19,20: manifolds of cathode gas discharge ports.

The overall model used in this paper is a PEMFC stack with 100 cells, which is assembled with clamping bolts. The model of single cell is shown in Figure 4 with BPPs, gas diffusion layers, proton exchange membrane and sealing gaskets. The clamping force is transmitted onto the endplate and the inside components of the fuel cell stack by ten bolts. Due to the symmetry of the structure, the model for static analysis uses a stack with 50 cells to simplify the numerical simulation, as shown in Figure 5. The simulations are carried out in software Ansys®.

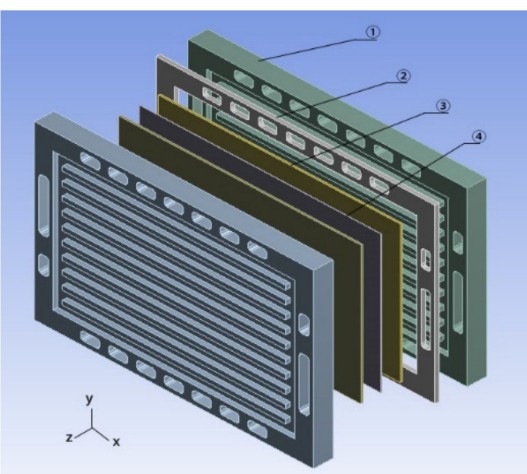

**Figure 4.** Single Cell Model ① Bipolar plate; ② Sealing gasket; ③ Gas diffusion layer; ④Proton exchange membrane.

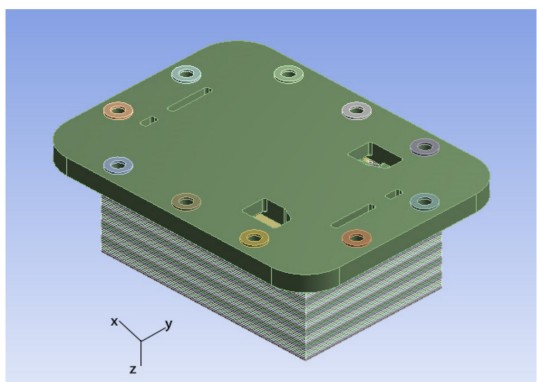

**Figure 5.** Fuel cell stack model.

The material parameters of the fuel cell stack are shown in Table 1, and the dimensional parameters are shown in Table 2.

**Table 1.** Material parameters of fuel cell components [23].

| Components | Material | Elastic Modulus/GPa | Poisson's Ratio | Density/kg/m³ |
|---|---|---|---|---|
| PEM | Nafion®112 | 0.32 | 0.4 | 500 |
| GDL | TGP-H-90 | 0.06 (thickness) 10 (in-plane) | 0.33 | 440 |
| Gasket | VMQ | 5.5 | 0.3 | 1700 |
| Bipolar plate | Graphite | 10 | 0.25 | 2160 |
| Current collector | Copper | 100 | 0.33 | 8940 |
| Insulator | POM | 2.6 | 0.386 | 1420 |
| Endplate | Aluminum alloy | 69 | 0.33 | 2800 |

**Table 2.** Dimensional parameters of fuel cell components.

| Components | Length/mm | Width/mm | Thickness/mm |
| --- | --- | --- | --- |
| PEM | 260 | 160 | 0.2 |
| GDL | 260 | 160 | 0.05 |
| Gasket | 300 (outer) 260 (inner) | 200 (outer) 160 (inner) | 0.45 |
| Bipolar plate | 300 | 200 | 2.09 |
| Current collector | 300 | 200 | 2 |
| Insulator | 300 | 200 | 3 |
| Endplate | 400 | 300 | 35 |
| Bolt gasket | D = 30 mm, d = 17 mm, H = 3 mm | | |

The clamping force is 30 kN, which is evenly distributed on ten bolts. The symmetry constraints are applied on the surface of bottom cell since the symmetric structure. The connection in the cell is set bonded. The connections between different cells are frictional and the frictional coefficient is 0.1 according to experience. The average mesh size is set to 5 mm. There are 1,015,575 nodes and 153,090 elements, and the evaluated average mesh quality is 0.75336.

For the intake endplate, the model used in topology optimization is shown in Figure 6, in which the green area is the design area. Considering the manufacturing process of aluminum alloy materials, a non-design area with a width of 7 mm is reserved at the edge of the endplate around the supply and discharge ports. The overall thickness of the end plate is 35 mm. In order to ensure the uniformity of stress and reduce the interference of subjective factors, 10 mm at the bottom (including the manifold area with a thickness of 5 mm) is reserved as the non-design area. The remaining part is the topology optimization design area.

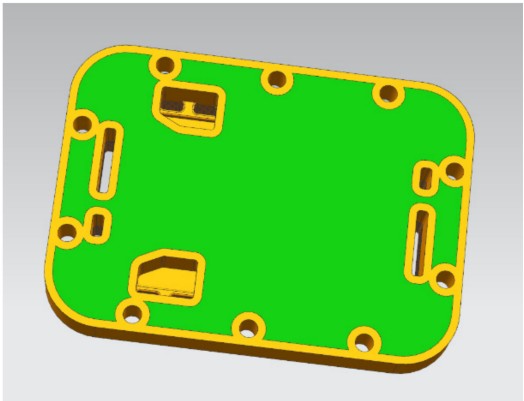

**Figure 6.** Model used in topology optimization.

The topology optimization is carried out in Optistruct. There are two optimization goals. One is to ensure that the endplate has sufficient stiffness, which is reflected as the minimum value of the strain energy. The function is expressed as:

$$\text{Min: } SE(\rho) \tag{1}$$

where $\rho$ represents the relative density of finite elements and SE represents the strain energy due to deformation.

The other goal is the uniform stress distribution of the first layer of single cells near to the endplate, which is reflected as the minimum value of the root mean square displacement of all nodes in the z-direction of the contact surface. The function is expressed as:

$$\text{Min: } RMS(\rho) \tag{2}$$

where RMS represents the root mean square.

Based on this, double-objective optimization makes the two goals coupled with one another. The function is expressed as:

$$\text{Min}: \left( \sqrt{p_1 * \frac{\text{SE}(\rho) - \text{SE}(\rho)_{\min}}{\text{SE}(\rho)_{\max} - \text{SE}(\rho)_{\min}}} + \sqrt{p_2 * \frac{\text{RMS}(\rho) - \text{RMS}(\rho)_{\min}}{\text{RMS}(\rho)_{\max} - \text{RMS}(\rho)_{\min}}} \right)^2 \quad (3)$$

where $\text{SE}(\rho)_{\min}$ and $\text{SE}(\rho)_{\max}$ are, respectively, the minimum and maximum strain energy. $\text{RMS}(\rho)_{\min}$ and $\text{RMS}(\rho)_{\max}$ are, respectively, the minimum and maximum root mean square. $p_1$ and $p_2$ are the weighting coefficients of the two goals. Here they are both set 0.5, which means the two goals are important in the same level. The boundary conditions are the same as the above simulation. The corresponding manufacturing constraints are also considered, which is in order to avoid the optimized results having no possibility of being manufactured. The minimum member size is used to control the minimum size of the reserved part of the topology optimization design results, aiming to control the checkerboard phenomenon and the degree of dispersion. Here it is set to 15 mm. The maximum member size prevents large or massive material aggregation in the result and forces a more discrete result; here it is set to 30 mm. The draft direction can be divided into one-way and two-way, aiming to eliminate unnecessary manufacturing design. It can also be used to generate the rib structure. In this paper, a single draft is preferred, and the direction is from the inside to the outside of the endplate.

## 3. Results and Discussions

### 3.1. Static Results of Endplates

Figure 7 is the stress distribution of the endplate obtained by the finite element simulation. The maximum stress is 12.711 MPa, the minimum stress is 0.0578 MPa and the average stress is 2.321 MPa. The maximum stress does not exceed the strength limit of the aluminum alloy, so there is no risk of damage. It can also be found that the stress gradually decreases from the bolt to the endplate center, but due to the existence of the supply, discharge ports and the manifold, the stress concentration is obvious. The maximum value of stress appears at the edge of the supply and discharge port, where the stress concentration effect is obvious, because it is close to the clamping bolts.

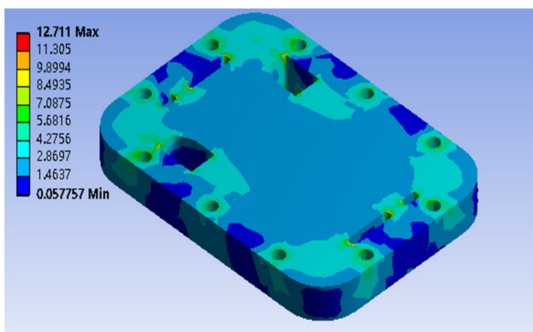

**Figure 7.** Stress distribution of intake endplate.

Figure 8 is the z-directional displacement of the endplate. The maximum displacement is 0.0177 mm, the minimum displacement is 0.00065 mm and the average displacement is 0.00496 mm. It can be found that when the endplate is under load, the outer sides are deformed greatly and slightly larger than those of the four corners. The deformation of the central part is relatively small. In other words, it presents a shape with a relatively convex center and a relatively concave surrounding.

Figure 9 is the stress distribution of the first cell next to the endplate. The maximum stress is 5.237 MPa, the minimum stress is 0.00544 MPa and the average stress is 0.662 MPa. The stress level of the sealing area is significantly higher than that of the flow channel area of fuel cell. In order to measure the inhomogeneity of the pressure distribution on this cell,

it is characterized by the coefficient of variation. The coefficient of variation is a normalized measure of the dispersion, which is defined as the ratio of the standard deviation to the mean. It is expressed as:

$$c_v = \frac{\sigma}{\mu} \tag{4}$$

where $c_v$ is the coefficient of variation, $\sigma$ is the standard deviation and $\mu$ is the mean. The standard deviation of the stress of the single cell in this layer is 0.963 MPa and the mean of the stress is 0.662 MPa, so the coefficient of variation is 1.454.

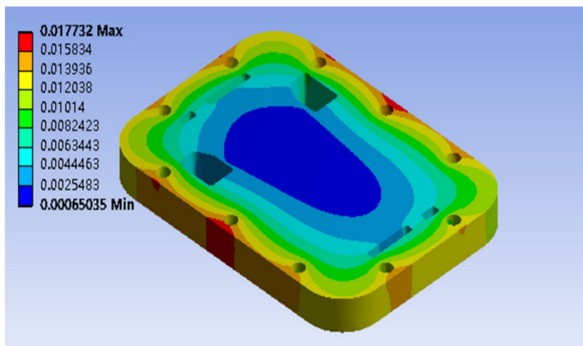

**Figure 8.** Z−directional deformation distribution of intake endplate.

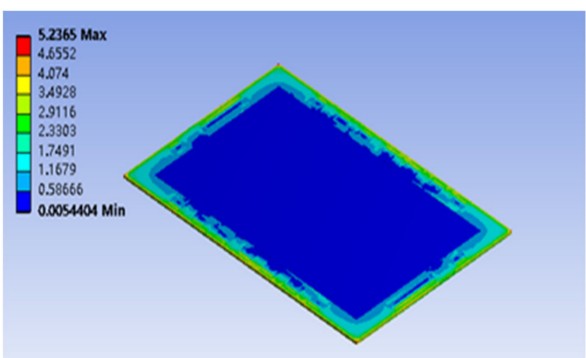

**Figure 9.** Stress distribution of first cell next to the intake endplate.

### 3.2. Topology Optimization

The optimization results show the relative density of the cells, which can be selected to be retained for larger densities and removed for smaller densities. Set a certain threshold in the software Hyperview®, and hide all the densities less than this value to leave the optimized model. Here, the threshold is set to 0.4. The specific results are shown in Figure 10. Different colors in the figure represent different density values of the endplates. The density of the red part is one, and the density of the dark blue part is close to zero.

As shown in Figure 10a, when it is aiming to reduce the strain energy and increase the stiffness of the endplates, the material is mainly concentrated around the clamping bolts and forms connections with the adjacent bolts. Obviously, this will increase the elastic deformation of the central part, thus leading to the uneven pressure distribution of the single cell. As shown in Figure 10b, in order to obtain a uniform pressure distribution, the material is mainly concentrated in the central part of the endplate, but this also obviously leads to large deformation and stress concentration of the endplate. As shown in Figure 10c, the optimization results of the two-objectives coupling can be roughly regarded as the combination of two single-objective optimization results, which have some characteristics of the those two.

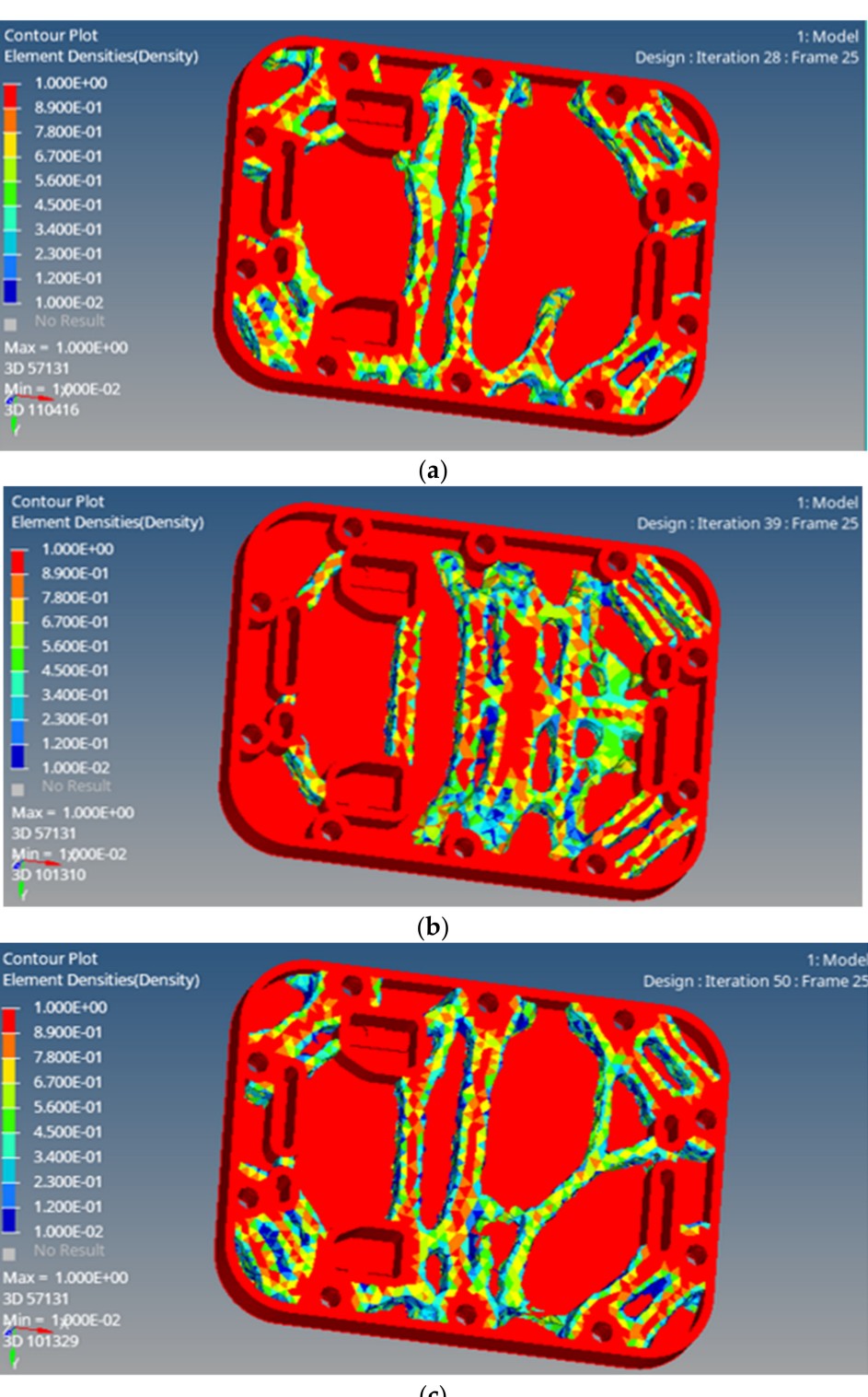

**Figure 10.** Intake endplate topology optimization with supply and discharge ports: (**a**) aim to minimize strain energy; (**b**) aim for uniform stress distribution; (**c**) two objectives coupling.

Since the blind endplate does not have structures such as supply and discharge ports, its topology-optimized structure is also significantly different to the intake endplate. Under the same boundary conditions, the optimization results are shown in Figure 11. Compared with the intake endplate, the material concentration is similar to the intake endplate. However, the topology structure is relatively regular and symmetric.

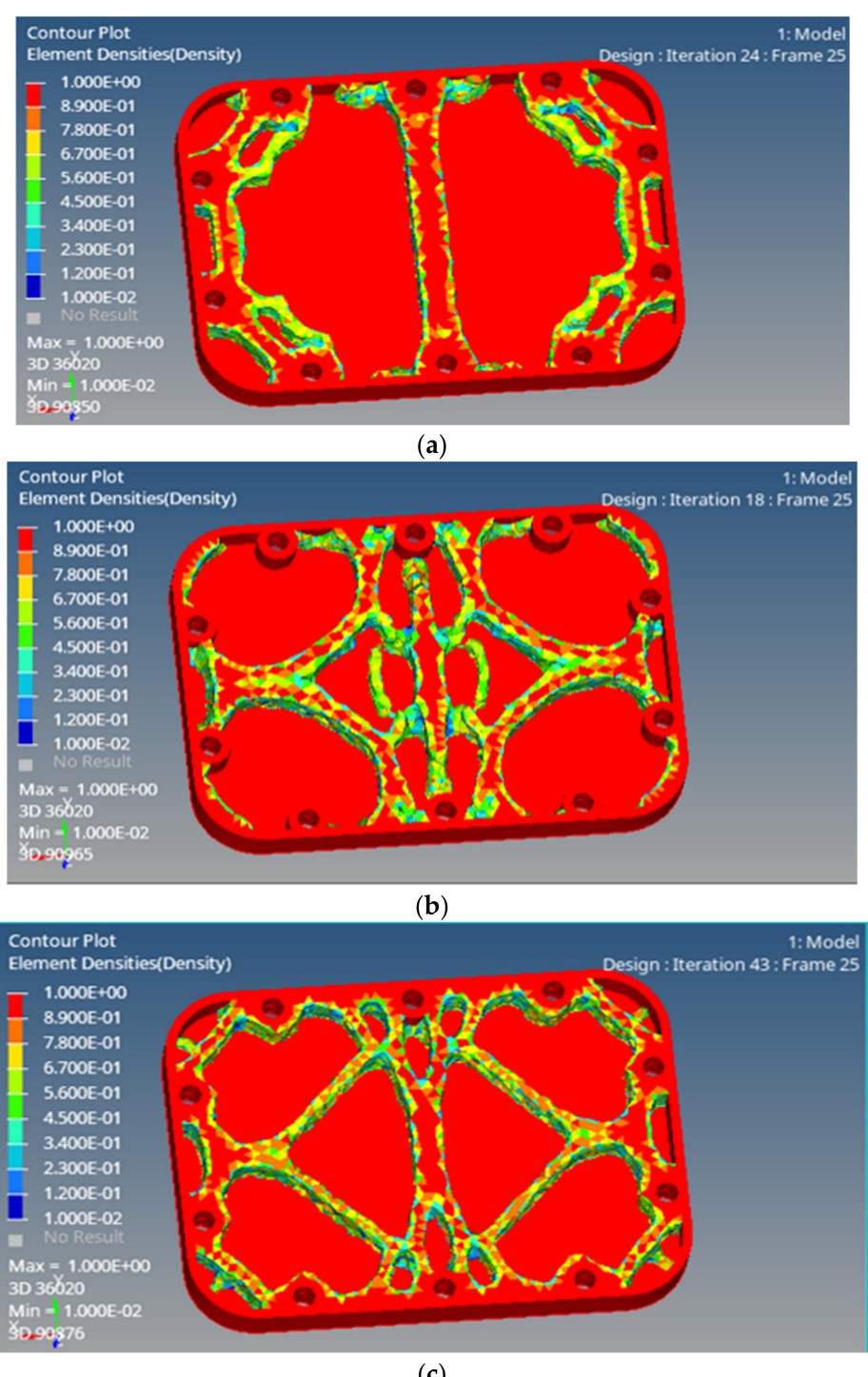

**Figure 11.** Blind endplate topology optimization with supply and discharge ports: (**a**) aim to minimize strain energy; (**b**) aim for uniform stress distribution; (**c**) two objectives coupling.

Because the optimized results only provide recommendations for material distribution, the results need to be carefully reconstructed and compared with the original endplates. Based on the topology-optimized structure of the intake endplate in Figure 10c, the geometrically reconstructed model is shown in Figure 12.

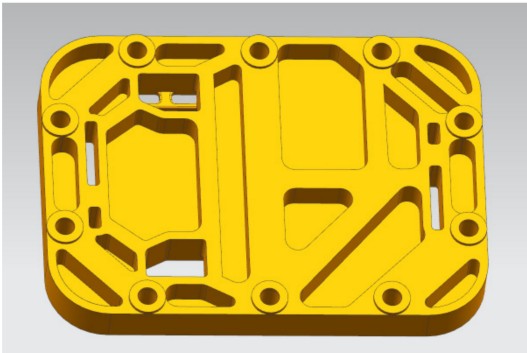

**Figure 12.** Reconstructed intake endplate.

In order to study the mechanical characteristics of the inlet endplate after topology optimization, the mechanical simulation is carried out. Figure 13 shows the stress distribution of the optimized endplate. The maximum stress of the endplate is 16.203 MPa, the minimum stress is 0.0354 MPa and the average stress is 2.783 MPa. The maximum stress is still far less than the strength limit of the aluminum alloy, and there is no risk of damage. The stress of the ribs between the adjacent bolts is relatively high, which shows the rationality of the topology optimization results. Figure 14 shows the z-directional deformation distribution of reconstructed intake endplate. Figure 15 shows the stress distribution of first cell next to the reconstructed intake endplate.

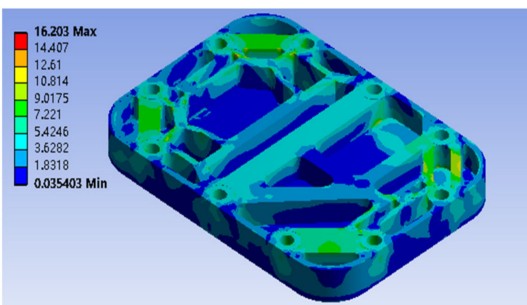

**Figure 13.** Stress distribution of reconstructed intake endplate.

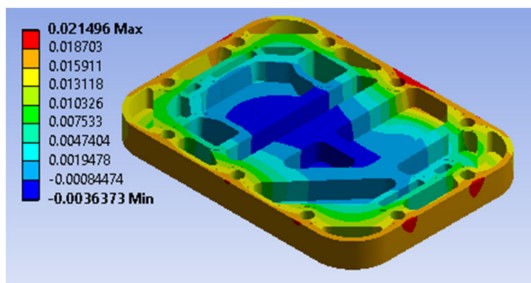

**Figure 14.** Z−directional deformation distribution of reconstructed intake endplate.

The differences between the topology-optimized intake endplate and the original intake endplate are shown in Table 3. Compared to the original endplate, the mass of the optimized intake endplate is reduced by 35%, thus effectively reducing the mass of the endplate for the goal of lightweight. On the other hand, the coefficient of variation of the first cell next to the endplate increases 1%, maintaining the stress uniformity after topology optimization.

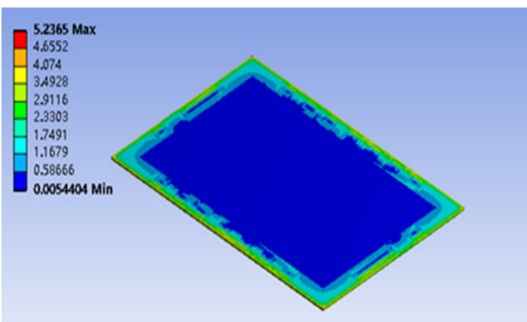

**Figure 15.** Stress distribution of first cell next to the reconstructed intake endplate.

**Table 3.** Differences between original and reconstructed intake endplate.

|  | Mass/kg | Maximum Stress/MPa | Average Stress/MPa | Coefficient of Variation of the First Cell Next to the Endplate |
|---|---|---|---|---|
| Original endplate | 8.019 | 12.711 | 2.321 | 1.454 |
| Optimized endplate | 5.103 | 16.203 | 2.783 | 1.467 |
| Change | Decrease 35% | Increase 27% | Increase 20% | Increase 1% |

Based on the topology-optimized structure in Figure 11c, the blind endplate after geometric reconstruction is shown in Figure 16.

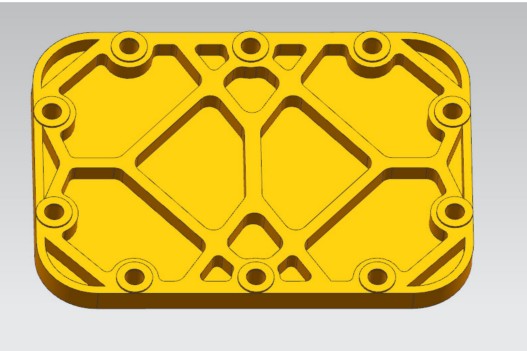

**Figure 16.** Reconstructed blind endplate.

The stress distribution of the optimized blind endplate is shown in Figure 17. The maximum stress is 19.009 MPa, the minimum stress is 0.093 MPa and the average stress is 3.506 MPa. The maximum stress is still far less than the strength limit of the aluminum alloy. Same as the reconstructed intake endplate, the rib stress between the adjacent bolts is also higher here.

The differences between the topology-optimized blind endplate and the original blind endplate are shown in Table 4. Compared to the original blind endplate, the mass of the optimized blind endplate is reduced by 46%, thus effectively reducing the mass of the end plate, achieving the goal of light weight. On the other hand, the stress value coefficient of variation of the first cell next to the endplate decreases 6%, which maintains stress uniformity of the blind endplate after topology optimization.

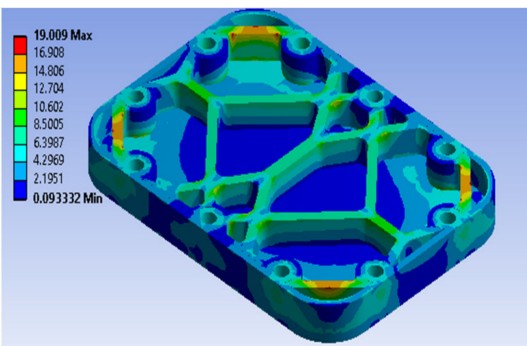

**Figure 17.** Stress distribution of reconstructed blind endplate.

**Table 4.** Differences between original blind endplate and reconstructed blind endplate.

|  | Mass/kg | Maximum Stress/MPa | Average Stress/MPa | Stress Value Coefficient of Variation of the First Single Cell Next to the Endplate |
|---|---|---|---|---|
| Original endplate | 8.451 | 5.855 | 1.556 | 1.274 |
| Optimized endplate | 4.590 | 19.009 | 3.506 | 1.353 |
| Change | Decrease 46% | Increase 224% | Increase 125% | Increase 6% |

## 4. Conclusions

In this paper, the endplate of fuel cell stacks, including the supply and discharge ports and distribution manifold structure, is firstly established by UG®, and the mechanical simulation is carried out by Ansys®. Then, the topology optimization is carried out, and the endplate after topology optimization is reconstructed, simulated and compared with the original endplate. The following conclusions are obtained:

- Under the clamping force, the endplate presents a shape with a relatively convex center and a relatively concave around the corners. The stresses at the supply and discharge ports are relatively large due to stress concentration.
- After the topology optimization of the endplate, the mass of the intake endplate can be reduced by 35%, and the mass of the blind endplate can be reduced by 46%, while maintaining the stress distribution uniformity of the endplates, the goal of a lightweight endplate is achieved.
- Due to the difference in structure between the intake endplate and the blind endplate, there are also obvious differences in topology results, which need to be designed separately.

The endplates after topology optimization obviously reduce the weight, and their strength also meets the requirements, which has certain reference significance for the design of endplates for fuel cell stacks.

**Author Contributions:** Conceptualization, Z.Z. and J.Z.; methodology, Z.Z.; software, J.Z.; validation, Z.Z. and J.Z.; resources, T.Z.; writing—original draft preparation, J.Z.; writing—review and editing, Z.Z.; project administration, Z.Z.; funding acquisition, Z.Z. All authors have read and agreed to the published version of the manuscript.

**Funding:** This research was funded by Natural & Science Foundation of Shanghai, grant number 22ZR1466800.

**Data Availability Statement:** Not applicable.

**Conflicts of Interest:** The authors declare no conflict of interest.

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
