# Peer review of "Endplate Design and Topology Optimization of Fuel Cell Stack Clamped with Bolts"

_sustainability, doi:10.3390/su14084730_

Round 1
Reviewer 1 Report
In this paper, a design and optimization are proposed for an end plate of for PEM fuel cell stacks with supply and discharge ports and distribution manifolds which are important to evaluate for optimization studies. Three optimized topologies aiming to minimize compliance, uniform stress distribution and two objectives coupling are discussed. The topology optimization results greatly reduced the mass of the endplate while keep certain stiffness, which is beneficial to get the lightweight endplates. With different optimization goals come different topology distributions. This study presents a different view for lightweight of fuel cell stacks, which has good engineering application significance. This manuscript is worth of publication. A few minor revisions are listed as followings:
- The title is clear and it is adequate to the content of the article.
- The novelty and originality of this work is also clear.
- The purposes or purported significance of the article is explicitly stated.
- The literature review and research study methods are explained clearly. Because this journal is a Science Citation Index Journal. In order to give our readers a sense of continuity, I can encourage you to identify journal publications of similar research in your papers. You should make a literature check of the papers published in recent years (2020, 2021 and even 2022).
- More details about the simulation settings are suggested be covered such as connection.
- The coefficient of variation and the topology objectives are better to be expressed by formula, which will be easier for readers to understand than description by words.
- The average stress column in Table 3 and Table 4 seems to be redundant. It is not an indicator of the damage risk or the uniformity.
- The conclusions are accurate and supported by the content.
- This manuscript can become well paper. If authors correct/enhance according to all whole of advices and recommendation, manuscript will become good level for publication in journal. These all the mistakes and errors can be corrected that I can accept this paper.
Author Response
Dear Reviewer,
Point 1: The title is clear and it is adequate to the content of the article.
Response 1: Thank you for your confirmation
Point 2: The novelty and originality of this work is also clear.
Response 2: Thank you for your confirmation
Point 3: The purposes or purported significance of the article is explicitly stated.
Response 3: Thank you for your confirmation
Point 4: The literature review and research study methods are explained clearly. Because this journal is a Science Citation Index Journal. In order to give our readers a sense of continuity, I can encourage you to identify journal publications of similar research in your papers. You should make a literature check of the papers published in recent years (2020, 2021 and even 2022).
Response 4: Some literatures are supplemented in the “Introduction” part, which are papers published in recent 5 years.
Point 5: More details about the simulation settings are suggested be covered such as connection.
Response 5: Details about connections are added in revised manuscript line 144-146 to make the progress more clear and more precise
Point 6: The coefficient of variation and the topology objectives are better to be expressed by formula, which will be easier for readers to understand than description by words.
Response 6: The formula is added in the manuscript (formula 1-4) to make it more understandable at the page 6.
Point 7: The average stress column in Table 3 and Table 4 seems to be redundant. It is not an indicator of the damage risk or the uniformity.
Response 7: Although the average stress isn’t an indicator of the damage risk or the uniformity, but it can reveal the stress difference of the whole endplate. This can’t be concluded from the maximum stress value. So it is prefered to be kept.
Point 8: The conclusions are accurate and supported by the content.
Response 8: Thank you for your confirmation
Point 9: This manuscript can become well paper. If authors correct/enhance according to all whole of advices and recommendation, manuscript will become good level for publication in journal. These all the mistakes and errors can be corrected that I can accept this paper.
Response 9: Thank you for your confirmation

Reviewer 2 Report
This paper presented topology optimization of endplate design for fuel cell stack. The optimized structure between the intake endplate and the blind endplate was compared. However, minor revision should be done.
- BPP and gasket in the fuel cell should be introduced in Section 2.
- On line 193, how did author obtain the standard deviation of the pressure?
- Figure 8 showed the stress distribution of first cell next to the endplate. However, there is no contact stress in the flow field, Why?
- In my opinion, the endplate deformation and stress distribution of first cell should be given after the optimization.
Author Response
Thank you very much for your comments!
Point 1: BPP and gasket in the fuel cell should be introduced in Section 2.
Response 1: The new figure 4 is added for the fuel cell with BPP, GDL, MEA and gasket in the page 4 and the structure of the single cell is introduced at line 128.
Point 2: On line 193, how did author obtain the standard deviation of the pressure?
Response 2: After the simulation, the targer vuale of certain nodes can be exported into one Excel file. The the standard deviation and the mean can be calculated with Excel or other tools. The mean value is added in the revised manuscript to make the caculation process more discernible.
Point 3: Figure 8 showed the stress distribution of first cell next to the endplate. However, there is no contact stress in the flow field, Why?
Response 3: Yes, there is stress in the flow field area. However, compared to the stress in the sealing part, the stress in the flow field area is relatively small and is not obvious in the figure 8. And some corresponding explicaiton are added in the revised manuscript at the line 222 in the page 7.
Point 4: In my opinion, the endplate deformation and stress distribution of first cell should be given after the optimization.
Response 4: Yes, followed as your suggestion, the endplate deformation and the stress distribution of first cell after the optimization have been added as figure 14 and figure 15 in revised manuscript as suggested. And some corresponding explicaiton are added in the revised manuscript from the line 277 to line 279 in the page 9.
